# M-Walk: Learning to Walk over Graphs using Monte Carlo Tree Search

*Yelong Shen[1], *Jianshu Chen[1], *Po-Sen Huang[2]⋆, Yuqing Guo[2], and Jianfeng Gao[2]

[1]Tencent AI Lab, Bellevue, WA, USA.
{yelongshen, jianshuchen}@tencent.com
[2]Microsoft Research, Redmond, WA, USA
{yuqguo, jfgao}@microsoft.com

## Abstract

Learning to walk over a graph towards a target node for a given query and a source node is an important problem in applications such as knowledge base completion (KBC). It can be formulated as a reinforcement learning (RL) problem with a known state transition model. To overcome the challenge of sparse rewards, we develop a graph-walking agent called M-Walk, which consists of a deep recurrent neural network (RNN) and Monte Carlo Tree Search (MCTS). The RNN encodes the state (i.e., history of the walked path) and maps it separately to a policy and Q-values. In order to effectively train the agent from sparse rewards, we combine MCTS with the neural policy to generate trajectories yielding more positive rewards. From these trajectories, the network is improved in an off-policy manner using Q-learning, which modifies the RNN policy via parameter sharing. Our proposed RL algorithm repeatedly applies this policy-improvement step to learn the model. At test time, MCTS is combined with the neural policy to predict the target node. Experimental results on several graph-walking benchmarks show that M-Walk is able to learn better policies than other RL-based methods, which are mainly based on policy gradients. M-Walk also outperforms traditional KBC baselines.

## 1 Introduction

We consider the problem of learning to walk over a graph in order to find a target node for a given source node and a query. Such problems appear in, for example, knowledge base completion (KBC) [38, 16, 31, 19, 7]. A knowledge graph is a structured representation of world knowledge in the form of entities and their relations (e.g., Figure 1(a)), and has a wide range of downstream applications such as question answering. Although a typical knowledge graph may contain millions of entities and billions of relations, it is usually far from complete. KBC aims to predict the missing relations between entities using information from the existing knowledge graph. More formally, let $\mathcal{G} = (\mathcal{N}, \mathcal{E})$ denote a graph, which consists of a set of nodes, $\mathcal{N} = \{n_i\}$, and a set of edges, $\mathcal{E} = \{e_{ij}\}$, that connect the nodes, and let $q$ denote an input query. The problem is stated as using the graph $\mathcal{G}$, the source node $n_S \in \mathcal{N}$ and the query $q$ as inputs to predict the target node $n_T \in \mathcal{N}$. In KBC tasks, $\mathcal{G}$ is a given knowledge graph, $\mathcal{N}$ is a collection of entities (nodes), and $\mathcal{E}$ is a set of relations (edges) that connect the entities. In the example in Figure 1(a), the objective of KBC is to identify the target node $n_T$ = USA for the given head entity $n_S$ = Obama and the given query $q$ = CITIZENSHIP.

The problem can also be understood as constructing a function $f(\mathcal{G}, n_S, q)$ to predict $n_T$, where the functional form of $f(\cdot)$ is generally unknown and has to be learned from a training dataset consisting of samples like $(n_S, q, n_T)$. In this work, we model $f(\mathcal{G}, n_S, q)$ by means of a graph-walking *agent* that intelligently navigates through a *subset* of nodes in the graph from $n_S$ towards $n_T$. Since $n_T$ is unknown, the problem cannot be solved by conventional search algorithms such as $A^*$-search [11], which seeks to find paths between the given source and target nodes. Instead, the *agent* needs to *learn* its search policy from the training dataset so that, after training is complete, the agent knows how to walk over the graph to reach the correct target node $n_T$ for an *unseen* pair of $(n_S, q)$. Moreover, each training sample is in the form of "(source node, query, target node)", and there is no intermediate supervision for the correct search path. Instead, the agent receives only delayed *evaluative* feedback: when the agent correctly (or incorrectly) predicts the target node in the training set, the agent will receive a positive (or zero) reward. For this reason, we formulate the problem as a Markov decision process (MDP) and train the agent by reinforcement learning (RL) [27].

The problem poses two major challenges. Firstly, since the state of the MDP is the entire trajectory, reaching a correct decision usually requires not just the query, but also the entire history of traversed nodes. For the KBC example in Figure 1(a), having access to the current node $n_t$ = Hawaii alone is not sufficient to know that the best action is moving to $n_{t+1}$ = USA. Instead, the agent must track the entire history, including the input query $q$ = Citizenship, to reach this decision. Secondly, the reward is sparse, being received only at the end of a search path, for instance, after correctly predicting $n_T$=USA.

In this paper, we develop a neural graph-walking agent, named *M-Walk*, that effectively addresses these two challenges. First, M-Walk uses a novel recurrent neural network (RNN) architecture to encode the entire history of the trajectory into a vector representation, which is further used to model the policy and the Q-function. Second, to address the challenge of sparse rewards, M-Walk exploits the fact that the MDP transition model is known and deterministic.[2] Specifically, it combines Monte Carlo Tree Search (MCTS) with the RNN to generate trajectories that obtain significantly more positive rewards than using the RNN policy alone. These trajectories can be viewed as being generated from an improved version of the RNN policy. But while these trajectories can improve the RNN policy, their off-policy nature prevents them from being leveraged by policy gradient RL methods. To solve this problem, we design a structure for sharing parameters between the Q-value network and the RNN's policy network. This allows the policy network to be indirectly improved through Q-learning over the off-policy trajectories. Our method is in sharp contrast to existing RL-based methods for KBC, which use a policy gradients (REINFORCE) method [36] and usually require a large number of rollouts to obtain a trajectory with a positive reward, especially in the early stages of learning [9, 37, 14]. Experimental results on several benchmarks, including a synthetic task and several real-world KBC tasks, show that our approach learns better policies than previous RL-based methods and traditional KBC methods.

The rest of the paper is organized as follows: Section 3 develops the M-Walk agent, including the model architecture, the training and testing algorithms.[3] Experimental results are presented in Section 4. Finally, we discuss related work in Section 5 and conclude the paper in Section 6.

## 2  Graph Walking as a Markov Decision Process

In this section, we formulate the graph-walking problem as a Markov Decision Process (MDP), which is defined by the tuple $(\mathcal{S}, \mathcal{A}, \mathcal{R}, \mathcal{P})$, where $\mathcal{S}$ is the set of states, $\mathcal{A}$ is the set of actions, $\mathcal{R}$ is the reward function, and $\mathcal{P}$ is the state transition probability. We further define $\mathcal{S}$, $\mathcal{A}$, $\mathcal{R}$ and $\mathcal{P}$ below. Figure 1(b) illustrates the MDP corresponding to the KBC example of Figure 1(a). Let $s_t \in \mathcal{S}$ denote the state at time $t$. Recalling that the agent needs the entire history of traversed nodes and the query to make a correct decision, we define $s_t$ by the following recursion:

$$s_t = s_{t-1} \cup \{a_{t-1}, n_t, \mathcal{E}_{n_t}, \mathcal{N}_{n_t}\}, \qquad s_0 \triangleq \{q, n_S, \mathcal{E}_{n_S}, \mathcal{N}_{n_S}\} \qquad (1)$$

where $a_t \in \mathcal{A}$ denotes the action selected by the agent at time $t$, $n_t \in \mathcal{G}$ denotes the currently visited node at time $t$, $\mathcal{E}_{n_t} \subset \mathcal{E}$ is the set of all edges connected to $n_t$, and $\mathcal{N}_{n_t} \subset \mathcal{N}$ is the set of all nodes

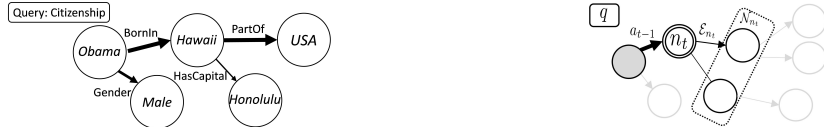

(a) An example of Knowledge Base Completion    (b) The corresponding Markov Decision Process

Figure 1: An example of Knowledge Base Completion and its formulation as a Markov Decision Process. (a) We want to identify the target node $n_T$ = USA for a given pair of query $q$ = Citizenship and source node $n_S$ = Obama. (b) The activated circles and edges (in black lines) denote all the observed information up to time $t$ (i.e., the state $s_t$). The double circle denotes the current node $n_t$, while $\mathcal{E}_{n_t}$ and $\mathcal{N}_{n_t}$ denote the edges and nodes connected to the current node.

connected to $n_t$ (i.e., the neighborhood). Note that state $s_t$ is a collection of (i) all the traversed nodes (along with their edges and neighborhoods) up to time $t$, (ii) all the previously selected (up to time $t-1$) actions, and (iii) the initial query $q$. The set $\mathcal{S}$ consists of all the possible values of $\{s_t, t \geq 0\}$. Based on $s_t$, the agent takes one of the following actions at each time $t$: (i) choosing an edge in $\mathcal{E}_{n_t}$ and moving to the next node $n_{t+1} \in \mathcal{N}_{n_t}$, or (ii) terminating the walk (denoted as the "STOP" action). Once the STOP action is selected, the MDP reaches the terminal state and outputs $\hat{n}_T = n_t$ as a prediction of the target node $n_T$. Therefore, we define the set of feasible actions at time $t$ as $\mathcal{A}_t \triangleq \mathcal{E}_{n_t} \cup \{\text{STOP}\}$, which is usually time-varying. The entire action space $\mathcal{A}$ is the union of all $\mathcal{A}_t$, i.e., $\mathcal{A} = \cup_t \mathcal{A}_t$. Recall that the training set consists of samples in the form of $(n_S, q, n_T)$. The reward is defined to be $+1$ when the predicted target node $\hat{n}_T$ is the same as $n_T$ (i.e., $\hat{n}_T = n_T$), and zero otherwise. In the example of Figure 1(a), for a training sample (Obama, Citizenship, USA), if the agent successfully navigates from Obama to USA and correctly predicts $\hat{n}_T$ = USA, the reward is $+1$. Otherwise, it will be $0$. The rewards are sparse because positive reward can be received only at the end of a correct path. Furthermore, since the graph $\mathcal{G}$ is known and static, the MDP transition probability $p(s_t|s_{t-1}, a_{t-1})$ is *known* and *deterministic*, and is defined by (1). To see this, we observe from Figure 1(b) that once an action $a_t$ (i.e., an edge in $\mathcal{E}_{n_t}$ or "STOP") is selected, the next node $n_{t+1}$ and its associated $\mathcal{E}_{n_{t+1}}$ and $\mathcal{N}_{n_{t+1}}$ are known. By (1) (with $t$ replaced by $t+1$), this means that the next state $s_{t+1}$ is determined. This important (model-based) knowledge will be exploited to overcome the sparse-reward problem using MCTS and significantly improve the performance of our method (see Sections 3–4 below).

We further define $\pi_\theta(a_t|s_t)$ and $Q_\theta(s_t, a_t)$ to be the policy and the Q-function, respectively, where $\theta$ is a set of model parameters. The policy $\pi_\theta(a_t|s_t)$ denotes the probability of taking action $a_t$ given the current state $s_t$. In M-Walk, it is used as a prior to bias the MCTS search. And $Q_\theta(s_t, a_t)$ defines the long-term reward of taking action $a_t$ at state $s_t$ and then following the optimal policy thereafter. The objective is to learn a policy that maximizes the terminal rewards, i.e., correctly identifies the target node with high probability. We now proceed to explain how to model and jointly learn $\pi_\theta$ and $Q_\theta$ to achieve this objective.

## 3 The M-Walk Agent

In this section, we develop a neural graph-walking agent named M-Walk (i.e., MCTS for graph Walking), which consists of (i) a novel neural architecture for jointly modeling $\pi_\theta$ and $Q_\theta$, and (ii) Monte Carlo Tree Search (MCTS). We first introduce the overall neural architecture and then explain how MCTS is used during the training and testing stages. Finally, we describe some further details of the neural architecture. Our discussion focuses on addressing the two challenges described earlier: history-dependent state and sparse rewards.

### 3.1 The neural architecture for jointly modeling $\pi_\theta$ and $Q_\theta$

Recall from Section 2 (e.g., (1)) that one challenge in applying RL to the graph-walking problem is that the state $s_t$ nominally includes the entire history of observations. To address this problem, we propose a special RNN encoding the state $s_t$ at each time $t$ into a vector representation, $h_t = \text{ENC}_{\theta_e}(s_t)$, where $\theta_e$ is the associated model parameter. We defer the discussion of this RNN state encoder to Section 3.4, and focus in this section on how to use $h_t$ to jointly model $\pi_\theta$ and

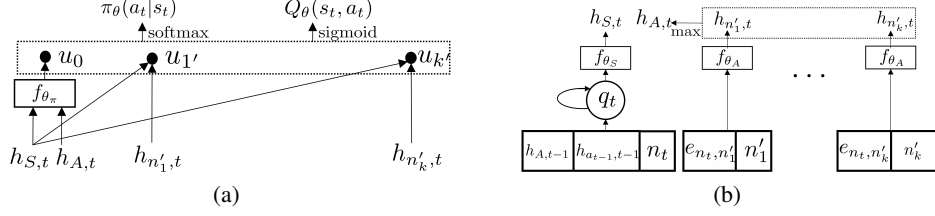

Figure 2: The neural architecture for M-Walk. (a) The vector representation of the state is mapped into $\pi_\theta$ and $Q_\theta$. (b) The GRU-RNN state encoder maps the state into its vector representation $h_t$. Note that the inputs $h_{A,t-1}$ and $h_{a_{t-1},t-1}$ are from the output of the previous time step $t-1$.

$Q_\theta$. Specifically, the vector $h_t$ consists of several sub-vectors of the same dimension $M$: $h_{S,t}$, $\{h_{n',t} : n' \in \mathcal{N}_{n_t}\}$ and $h_{A,t}$. Each sub-vector encodes part of the state $s_t$ in (1). For instance, the vector $h_{S,t}$ encodes $(s_{t-1}, a_{t-1}, n_t)$, which characterizes the history in the state. The vector $h_{n',t}$ encodes the (neighboring) node $n'$ and the edge $e_{n_t,n'}$ connected to $n_t$, which can be viewed as a vector representation of the $n'$-th candidate action (excluding the STOP action). And the vector $h_{A,t}$ is a vector summarization of $\mathcal{E}_{n_t}$ and $\mathcal{N}_{n_t}$, which is used to model the STOP action probability. In summary, we use the sub-vectors to model $\pi_\theta$ and $Q_\theta$ according to:

$$u_0 = f_{\theta_\pi}(h_{S,t}, h_{A,t}), \quad u_{n'} = \langle h_{S,t}, h_{n',t} \rangle, \quad n' \in \mathcal{N}_{n_t} \tag{2}$$

$$Q_\theta(s_t, \cdot) = \sigma(u_0, u_{n'_1}, \ldots, u_{n'_k}), \ \pi_\theta(\cdot|s_t) = \phi_\tau(u_0, u_{n'_1}, \ldots, u_{n'_k}) \tag{3}$$

where $\langle \cdot, \cdot \rangle$ denotes inner product, $f_{\theta_\pi}(\cdot)$ is a fully-connected neural network with model parameter $\theta_\pi$, $\sigma(\cdot)$ denotes the element-wise sigmoid function, and $\phi_\tau(\cdot)$ is the softmax function with temperature parameter $\tau$. Note that we use the inner product between the vectors $h_{S,t}$ and $h_{n',t}$ to compute the (pre-softmax) score $u_{n'}$ for choosing the $n'$-th candidate action, where $n' \in \mathcal{N}_{n_t}$. The inner product operation has been shown to be useful in modeling Q-functions when the candidate actions are described by vector representations [13, 3] and in solving other problems [33, 1]. Moreover, the value of $u_0$ is computed by $f_{\theta_\pi}(\cdot)$ using $h_{S,t}$ and $h_{A,t}$, where $u_0$ gives the (pre-softmax) score for choosing the STOP action. We model the Q-function by applying element-wise sigmoid to $u_0, u_{n'_1}, \ldots, u_{n'_k}$, and we model the policy by applying the softmax operation to the *same* set of $u_0, u_{n'_1}, \ldots, u_{n'_k}$.[4] Note that the policy network and the Q-network share the same set of model parameters. We will explain in Section 3.2 how such parameter sharing enables indirect updates to the policy $\pi_\theta$ via Q-learning from off-policy data.

## 3.2 The training algorithm

We now discuss how to train the model parameters $\theta$ (including $\theta_\pi$ and $\theta_e$) from a training dataset $\{(n_S, q, n_T)\}$ using reinforcement learning. One approach is the policy gradient method (RE-INFORCE) [36, 28], which uses the current policy $\pi_\theta(a_t|s_t)$ to roll out multiple trajectories $(s_0, a_0, r_0, s_1, \ldots)$ to estimate a stochastic gradient, and then updates the policy $\pi_\theta$ via stochastic gradient ascent. Previous RL-based KBC methods [38, 5] typically use REINFORCE to learn the policy. However, policy gradient methods generally suffer from low sample efficiency, especially when the reward signal is sparse, because large numbers of Monte Carlo rollouts are usually needed to obtain many trajectories with positive terminal reward, particularly in the early stages of learning. To address this challenge, we develop a novel RL algorithm that uses MCTS to exploit the deterministic MDP transition defined in (1). Specifically, on each MCTS simulation, a trajectory is rolled out by selecting actions according to a variant of the PUCT algorithm [21, 25] from the root state $s_0$ (defined in (1)):

$$a_t = \mathrm{argmax}_a \left\{ c \cdot \pi_\theta(a|s_t)^\beta \sqrt{\sum_{a'} N(s_t, a')} / (1 + N(s_t, a)) + W(s_t, a) / N(s_t, a) \right\} \tag{4}$$

where $\pi_\theta(a|s)$ is the policy defined in Section 3.1, $c$ and $\beta$ are two constants that control the level of exploration, and $N(s, a)$ and $W(s, a)$ are the visit count and the total action reward accumulated on the $(s, a)$-th edge on the MCTS tree. Overall, PUCT treats $\pi_\theta$ as a prior probability to bias the MCTS

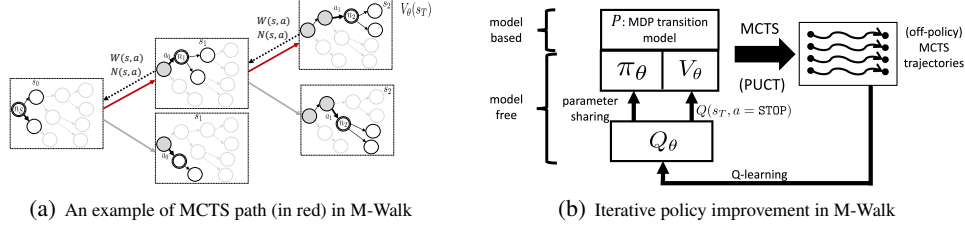

(a) An example of MCTS path (in red) in M-Walk

(b) Iterative policy improvement in M-Walk

Figure 3: MCTS is used to generate trajectories for iterative policy improvement in M-Walk.

search; PUCT initially prefers actions with high values of $\pi_\theta$ and low visit count $N(s, a)$ (because the first term in (4) is large), but then asympotically prefers actions with high value (because the first term in (4) vanishes and the second term $W(s, a)/N(s, a)$ dominates). When PUCT selects the STOP action or the maximum search horizon has been reached, MCTS completes one simulation and updates $W(s, a)$ and $N(s, a)$ using $V_\theta(s_T) = Q_\theta(s_T, a = \text{STOP})$. (See Figure 3(a) for an example and Appendix B.1 for more details.) The key idea of our method is that running multiple MCTS simulations generates a set of trajectories with more positive rewards (see Section 4 for more analysis), which can also be viewed as being generated by an improved policy $\pi_\theta$. Therefore, learning from these trajectories can further improve $\pi_\theta$. Our RL algorithm repeatedly applies this policy-improvement step to refine the policy. However, since these trajectories are generated by a policy that is different from $\pi_\theta$, they are off-policy data, breaking the assumptions inherent in policy gradient methods. For this reason, we instead update the Q-network from these trajectories in an off-policy manner using Q-learning: $\theta \leftarrow \theta + \alpha \cdot \nabla_\theta Q_\theta(s_t, a_t) \times (r(s_t, a_t) + \gamma \max_{a'} Q_\theta(s_{t+1}, a') - Q_\theta(s_t, a_t))$. Recall from Section 3.1 that $\pi_\theta$ and $Q_\theta(s, a)$ share the same set of model parameters; once the Q-network is updated, the policy network $\pi_\theta$ will also be automatically improved. Finally, the new $\pi_\theta$ is used to control the MCTS in the next iteration. The main idea of the training algorithm is summarized in Figure 3(b).

### 3.3 The prediction algorithm

At test time, we want to infer the target node $n_T$ for an unseen pair of $(n_S, q)$. One approach is to use the learned policy $\pi_\theta$ to walk through the graph $\mathcal{G}$ to find $n_T$. However, this would not exploit the known MDP transition model (1). Instead, we combine the learned $\pi_\theta$ and $Q_\theta$ with MCTS to generate an MCTS search tree, as in the training stage. Note that there could be multiple paths that reach the same terminal node $n \in \mathcal{G}$, meaning that there could be multiple leaf states in MCTS corresponding to that node. Therefore, the prediction results from these MCTS leaf states need to be merged into one score to rank the node $n$. Specifically, we use $\text{Score}(n) = \sum_{s_T \to n} N(s_T, a_T)/N \times Q_\theta(s_T, \text{STOP})$, where $N$ is the total number of MCTS simulations, and the summation is over all the leaf states $s_T$ that correspond to the same node $n \in \mathcal{G}$. $\text{Score}(n)$ is a weighted average of the terminal state values associated with the same candidate node $n$.[5] Among all the candidates nodes, we select the predicted target node to be the one with the highest score: $\hat{n}_T = \arg\max_n \text{Score}(n)$.

### 3.4 The RNN state encoder

We now discuss the details of the RNN state encoder $h_t = \text{ENC}_{\theta_e}(s_t)$, where $\theta_e \triangleq \{\theta_A, \theta_S, \theta_q\}$, as shown in Figure 2(b). Specifically, we explain how the sub-vectors of $h_t$ are computed. We introduce $q_t \triangleq s_{t-1} \cup \{a_{t-1}, n_t\}$ as an auxiliary variable. Then, the state $s_t$ in (1) can be written as $s_t = q_t \cup \{\mathcal{E}_{n_t}, \mathcal{N}_{n_t}\}$. Note that the state $s_t$ is composed of two parts: (i) $\mathcal{E}_{n_t}$ and $\mathcal{N}_{n_t}$, which represent the candidate actions to be selected (excluding the STOP action), and (ii) $q_t$, which represents the history. We use two different neural networks to encode these separately. For the $n'$-th candidate action ($n' \in \mathcal{N}_{n_t}$), we concatenate $n'$ with its associated $e_{n_t, n'} \in \mathcal{E}_{n_t}$ and input them into a fully connected network (FCN) $f_{\theta_A}(\cdot)$ to compute their joint vector representation $h_{n', t}$, where $\theta_A$ is the model parameter. Recall that the action space $\mathcal{A}_t = \mathcal{E}_{n_t} \cup \{\text{STOP}\}$ can be time-varying when the size of $\mathcal{E}_{n_t}$ changes over time. To address this issue, we apply the same FCN $f_\theta(\cdot)$ to

different $(n', e_{n_t,n'})$ to obtain their respective representations. Then, we use a coordinate-wise max-pooling operation over $\{h_{n',t} : n' \in \mathcal{N}_{n_t}\}$ to obtain a (fixed-length) overall vector representation of $\{\mathcal{E}_{n_t}, \mathcal{N}_{n_t}\}$. To encode $q_t$, we call upon the following recursion for $q_t$ (see Appendix A for the derivation): $q_{t+1} = q_t \cup \{\mathcal{E}_{n_t}, \mathcal{N}_{n_t}, a_t, n_{t+1}\}$. Inspired by this recursion, we propose using the GRU-RNN [4] to encode $q_t$ into a vector representation[6]: $q_{t+1} = f_{\theta_q}(q_t, [h_{A,t}, h_{a_t,t}, n_{t+1}])$ with initialization $q_0 = f_{\theta_q}(q, [0, 0, n_S])$, where $\theta_q$ is the model parameter, and $h_{a_t,t}$ denotes the vector $h_{n',t}$ at $n' = a_t$. We use $h_{A,t}$ and $h_{a_t,t}$ computed by the FCNs to represent $(\mathcal{E}_{n_t}, \mathcal{N}_{n_t})$ and $a_t$, respectively. Then, we map $q_t$ to $h_{S,t}$ using another FCN $f_{\theta_S}(\cdot)$.

## 4 Experiments

We evaluate and analyze the effectiveness of M-Walk on a synthetic Three Glass Puzzle task and two real-world KBC tasks. We briefly describe the tasks here, and give the experiment details and hyperparameters in Appendix B.

**Three Glass Puzzle** The Three Glass Puzzle [20] is a problem studied in math puzzles and graph theory. It involves three milk containers $\mathcal{A}$, $\mathcal{B}$, and $\mathcal{C}$, with capacities $A$, $B$ and $C$ liters, respectively. The containers display no intermediate markings. There are three feasible actions at each time step: (i) *fill* a container (to its capacity), (ii) *empty* all of its liquid, and (iii) *pour* its liquid into another container (up to its capacity). The objective of the problem is, given a desired volume $q$, to take a sequence of actions on the three containers after which one of them contains $q$ liters of liquid. We formulate this as a graph-walking problem; in the graph $\mathcal{G}$, each node $n = (a, b, c)$ denotes the amounts of remaining liquid in the three containers, each edge denotes one of the three feasible actions, and the input query is the desired volume $q$. The reward is $+1$ when the agent successfully fills one of the containers to $q$ and 0 otherwise (see Appendix B.2.1 for the details). We use vanilla policy gradient (REINFORCE) [36] as the baseline, with task success rate as the evaluation metric.

**Knowledge Base Completion** We use WN18RR and NELL995 knowledge graph datasets for evaluation. WN18RR [6] is created from the original WN18 [2] by removing various sources of test leakage, making the dataset more challenging. The NELL995 dataset was released by [38] and has separate graphs for each query relation. We use the same data split and preprocessing protocol as in [6] for WN18RR and in [38, 5] for NELL995. As in [38, 5], we study the 10 relation tasks of NELL995 separately. We use HITS@1,3 and mean reciprocal rank (MRR) as the evaluation metrics for WN18RR, and use mean average precision (MAP) for NELL995,[7] where HITS@$K$ computes the percentage of the desired entities being ranked among the top-$K$ list, and MRR computes an average of the reciprocal rank of the desired entities. We compare against RL-based methods [38, 5], embedding-based models (including DistMult [39], ComplEx [32] and ConvE [6]) and recent work in logical rules (NeuralLP) [40]. For all the baseline methods, we used the implementation released by the corresponding authors with their best-reported hyperparameter settings.[8] The details of the hyperparameters for M-Walk are described in Appendix B.2.2 of the supplementary material.

### 4.1 Performance of M-Walk

We first report the overall performance of the M-Walk algorithm on the three tasks and compare it with other baseline methods. We ran the experiments three times and report the means and standard deviations (except for PRA, TransE, and TransR on NELL995, whose results are directly quoted from [38]). On the Three Glass Puzzle task, M-Walk significantly outperforms the baseline: the best model of M-Walk achieves an accuracy of $(99.0 \pm 1.0)\%$ while the best REINFORCE method achieves $(49.0 \pm 2.6)\%$ (see Appendix C for more experiments with different settings on this task). For the two KBC tasks, we report their results in Tables 1-2, where PG-Walk and Q-Walk are two methods we created *just for the ablation study in the next section*. The proposed method outperforms previous works in most of the metrics on NELL995 and WN18RR datasets. Additional experiments on the FB15k-237 dataset can be found in Appendix C.1.1 of the supplementary material.

Table 1: The MAP scores (%) on NELL995 task, where we report RL-based methods in terms of "mean (standard deviation)". PG-Walk and Q-Walk are methods we created just for the ablation study.

| Tasks | M-Walk | PG-Walk | Q-Walk | MINERVA | DeepPath | PRA | TransE | TransR |
|---|---|---|---|---|---|---|---|---|
| AthletePlaysForTeam | **84.7** (1.3) | 80.8 (0.9) | 82.6 (1.2) | 82.7 (0.8) | 72.1 (1.2) | 54.7 | 62.7 | 67.3 |
| AthletePlaysInLeague | **97.8** (0.2) | 96.0 (0.6) | 96.2 (0.8) | 95.2 (0.8) | 92.7 (5.3) | 84.1 | 77.3 | 91.2 |
| AthleteHomeStadium | 91.9 (0.1) | 91.9 (0.3) | 91.1 (1.3) | **92.8** (0.1) | 84.6 (0.8) | 85.9 | 71.8 | 72.2 |
| AthletePlaysSport | 98.3 (0.1) | 98.0 (0.8) | 97.0 (0.2) | **98.6** (0.1) | 91.7 (4.1) | 47.4 | 87.6 | 96.3 |
| TeamPlaySports | **88.4** (1.8) | 87.4 (0.9) | 78.5 (0.6) | 87.5 (0.5) | 69.6 (6.7) | 79.1 | 76.1 | 81.4 |
| OrgHeadquaterCity | **95.0** (0.7) | 94.0 (0.4) | 94.0 (0.6) | 94.5 (0.3) | 79.0 (0.0) | 81.1 | 62.0 | 65.7 |
| WorksFor | **84.2** (0.6) | 84.0 (1.6) | 82.7 (0.2) | 82.7 (0.5) | 69.9 (0.3) | 68.1 | 67.7 | 69.2 |
| BornLocation | 81.2 (0.0) | **82.3** (0.6) | 81.4 (0.5) | 78.2 (0.0) | 75.5 (0.5) | 66.8 | 71.2 | 81.2 |
| PersonLeadsOrg | **88.8** (0.5) | 87.2 (0.5) | 86.9 (0.5) | 83.0 (2.6) | 79.0 (1.0) | 70.0 | 75.1 | 77.2 |
| OrgHiredPerson | **88.8** (0.6) | 87.2 (0.4) | 87.8 (0.9) | 87.0 (0.3) | 73.8 (1.9) | 59.9 | 71.9 | 73.7 |
| Overall | **89.9** | 88.9 | 87.8 | 87.6 | 78.8 | 69.7 | 72.3 | 77.5 |

Table 2: The results on the WN18RR dataset, in the form of "mean (standard deviation)".

| Metric (%) | M-Walk | PG-Walk | Q-Walk | MINERVA | ComplEx | ConvE | DistMult | NeuralLP |
|---|---|---|---|---|---|---|---|---|
| HITS@1 | **41.4** (0.1) | 39.3 (0.2) | 38.2 (0.3) | 35.1 (0.1) | 38.5 (0.3) | 39.6 (0.3) | 38.4 (0.4) | 37.2 (0.1) |
| HITS@3 | 44.5 (0.2) | 41.9 (0.1) | 40.8 (0.4) | 44.5 (0.4) | 43.9 (0.3) | **44.7** (0.2) | 42.4 (0.3) | 43.4 (0.1) |
| MRR | **43.7 (0.1)** | 41.3 (0.1) | 40.1 (0.3) | 40.9 (0.1) | 42.2 (0.2) | 43.3 (0.2) | 41.3 (0.3) | 43.5 (0.1) |

## 4.2 Analysis of M-Walk

We performed extensive experimental analysis to understand the proposed M-Walk algorithm, including (i) the contributions of different components, (ii) its ability to overcome sparse rewards, (iii) hyperparameter analysis, (iv) its strengths and weaknesses compared to traditional KBC methods, and (v) its running time. First, we used ablation studies to analyze the contributions of different components in M-Walk. To understand the contribution of the proposed neural architecture in M-Walk, we created a method, PG-Walk, which uses the same neural architecture as M-Walk but with the same training (PG) and testing (beam search) algorithms as MINERVA [5]. We observed that the novel neural architecture of M-Walk contributes an overall 1% gain relative to MINERVA on NELL995, and it is still 1% worse than M-Walk, which uses MCTS for training and testing. To further understand the contribution of MCTS, we created another method, Q-Walk, which uses the same model architecture as M-Walk except that it is trained by Q-learning only without MCTS. Note that this lost about 2% in overall performance on NELL995. We observed similar trends on WN18RR. In addition, we also analyze the importance of MCTS in the testing stage in Appendix C.1.

Second, we analyze the ability of M-Walk to overcome the sparse-reward problem. In Figure 4, we show the positive reward rate (i.e., the percentage of trajectories with positive reward during training) on the Three Glass Puzzle task and the NELL995 tasks. Compared to the policy gradient method (PG-Walk), and Q-learning method (Q-Walk) methods under the same model architecture, M-Walk with MCTS is able to generate trajectories with more positive rewards, and this continues to improve as training progresses. This confirms our motivation of using MCTS to generate higher-quality trajectories to alleviate the sparse-reward problem in graph walking.

Third, we analyze the performance of M-Walk under different numbers of MCTS rollout simulations and different search horizons on WN18RR dataset, with results shown in Figure 5(a). We observe that the model is less sensitive to search horizon and more sensitive to the number of MCTS rollouts. Finally, we analyze the strengths and weaknesses of M-Walk relative to traditional methods on the WN18RR dataset. The first question is how M-Walk performs on reasoning paths of different lengths compared to baselines. To answer this, we analyze the HITS@1 accuracy against ConvE in Fig. 5(b). We categorize each test example using the BFS (breadth-first search) steps from the query entity to the target entity (-1 means not reachable). We observe that M-Walk outperforms the strong baseline ConvE by 4.6–10.9% in samples that require 2 or 3 steps, while it is nearly on par for paths of length one. Therefore, M-Walk does better at reasoning over longer paths than ConvE. Another question is what are the major types of errors made by M-Walk. Recall that M-Walk only walks through a subset of the graph and ranks a subset of candidate nodes (e.g., MCTS produces about 20–60 unique candidates on WN18RR). When the ground truth is not in the candidate set, M-Walk always makes mistakes and we define this type of error as *out-of-candidate-set error*. To examine this effect, we show in Figure 5(c)-top the HITS@K accuracies when the ground truth is in the candidate

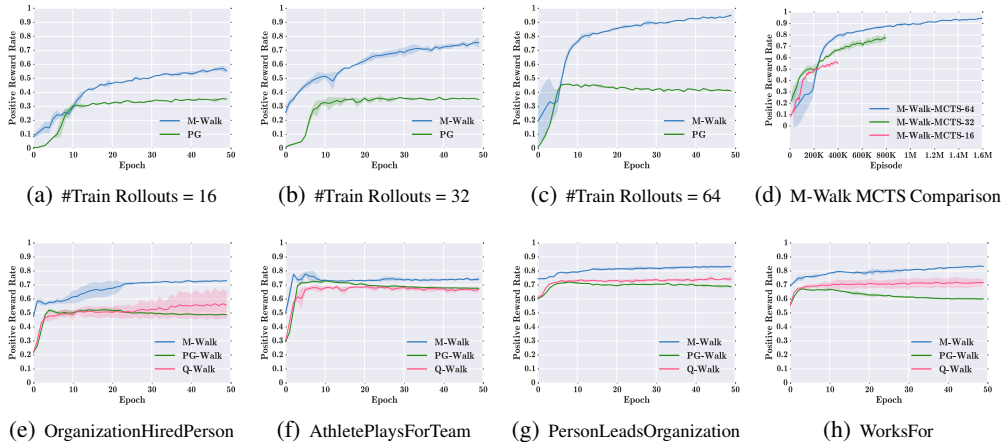

(a) #Train Rollouts = 16    (b) #Train Rollouts = 32    (c) #Train Rollouts = 64    (d) M-Walk MCTS Comparison

(e) OrganizationHiredPerson    (f) AthletePlaysForTeam    (g) PersonLeadsOrganization    (h) WorksFor

Figure 4: The positive reward rate. Figures (a)-(d) are the results on the Three Glass Puzzle task and Figures (e)-(h) are the results on the NELL-995 task. (See Appendix C.1.1 for more results.)

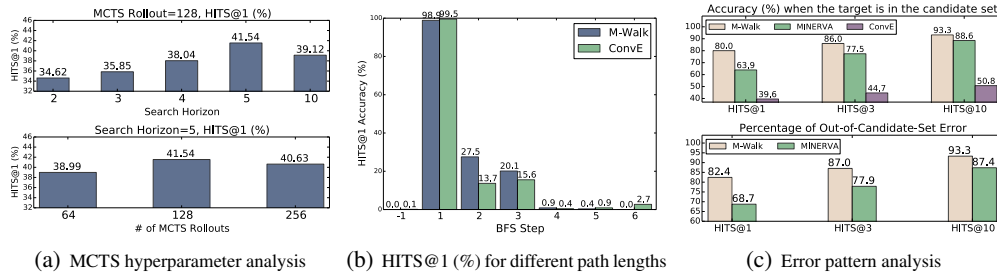

(a) MCTS hyperparameter analysis    (b) HITS@1 (%) for different path lengths    (c) Error pattern analysis

Figure 5: M-Walk hyperparameter and error analysis on WN18RR.

set.[9] It shows that M-Walk has very high accuracy in this case, which is significantly higher than ConvE (80% vs 39.6% in HITS@1). We further examine the percentage of out-of-candidate-set errors among all errors in Figure 5(c)-bottom. It shows that the major error made by M-Walk is the out-of-candidate-set error. These observations point to an important direction for improving M-Walk in future work: increasing the chance of covering the target by the candidate set.

Table 3: Running time of M-Walk and MINERVA for different combinations of (horizon, rollouts).

| Model | M-Walk (5,64) | M-Walk (5,128) | M-Walk (3,64) | M-Walk (3,128) | MINERVA (3,100), best |
|---|---|---|---|---|---|
| Training (hrs.) | 8 | 14 | 5 | 8 | 3 |
| Testing (sec/sample) | $3 \times 10^{-3}$ | $6 \times 10^{-3}$ | $1.6 \times 10^{-3}$ | $2.7 \times 10^{-3}$ | $2 \times 10^{-2}$ |

In Table 3, we show the running time of M-Walk (in-house C++ & Cuda) and MINERVA (TensorFlow-gpu) for both training and testing on WN18RR with different values of search horizon and number of rollouts (or MCTS simulation number). Note that the running time of M-Walk is comparable to that of MINERVA. Additional results can be found in Figure 9(c) of the supplementary material. Finally, in Table 4, we show examples of reasoning paths found by M-Walk.[10]

## 5 Related Work

**Reinforcement Learning** Recently, deep reinforcement learning has achieved great success in many artificial intelligence problems [17, 24, 25]. The use of deep neural networks with RL allows policies to be learned from raw data (e.g., images) in an end-to-end manner. Our work also aligns

Table 4: Examples of reasoning paths found by M-Walk on the NELL-995 dataset for the relation "AthleteHomeStadium". True (False) means the prediction is correct (wrong).

---

**AthleteHomeStadium:**

*Example 1*: athlete ernie banks $\xrightarrow{\text{AthleteHomeStadium}}$?

athlete ernie banks $\xrightarrow{\text{AthletePlaysInLeague}}$ SportsLeague mlb $\xrightarrow{\text{TeamPlaysInLeague}^{-1}}$ SportsTeam chicago cubs $\xrightarrow{\text{TeamHomeStadium}}$ StadiumOrEventVenue wrigley field, (True)

*Example 2*: coach jim zorn $\xrightarrow{\text{AthleteHomeStadium}}$?

coach jim zorn $\xrightarrow{\text{CoachWonTrophy}}$ AwardTrophyTournament super bowl $\xrightarrow{\text{TeamWonTrophy}^{-1}}$ SportsTeam redskins $\xrightarrow{\text{TeamHomeStadium}}$ StadiumOrEventVenue fedex field, (True)

*Example 3*: athlete oliver perez $\xrightarrow{\text{AthleteHomeStadium}}$?

athlete oliver perez $\xrightarrow{\text{AthletePlaysInLeague}}$ SportsLeague mlb $\xrightarrow{\text{TeamPlaysInLeague}^{-1}}$ SportsTeam chicago cubs $\xrightarrow{\text{TeamHomeStadium}}$ StadiumOrEventVenue wrigley field, (False)

---

with this direction. Furthermore, the idea of using an RNN to encode the history of observations also appeared in [12, 35]. The combination of model-based and model-free information in our work shares the same spirit as [24, 25, 26, 34]. Among them, the most relevant are [24, 25], which combine MCTS with neural policy and value functions to achieve superhuman performance on Go. Different from our work, the policy and the value networks in [24] are trained separately without the help of MCTS, and are only used to help MCTS after being trained. The work [25] uses a new policy iteration method that combines the neural policy and value functions with MCTS during training. However, the method in [25] improves the policy network from the MCTS probabilities of the moves, while our method improves the policy from the trajectories generated by MCTS. Note that the former is constructed from the visit counts of all the edges connected to the MCTS root node; it only uses information near the root node to improve the policy. By contrast, we improve the policy by learning from the trajectories generated by MCTS, using information over the entire MCTS search tree.

**Knowledge Base Completion** In KBC tasks, early work [2] focused on learning vector representations of entities and relations. Recent approaches have demonstrated limitations of these prior approaches: they suffer from cascading errors when dealing with compositional (multi-step) relationships [10]. Hence, recent works [8, 18, 10, 15, 30] have proposed approaches for injecting multi-step paths such as random walks through sequences of triples during training, further improving performance on KBC tasks. IRN [23] and Neural LP [40] explore multi-step relations by using an RNN controller with attention over an external memory. Compared to RL-based approaches, it is hard to interpret the traversal paths, and these models can be computationally expensive to access the entire graph in memory [23]. Two recent works, DeepPath [38] and MINERVA [5], use RL-based approaches to explore paths in knowledge graphs. DeepPath requires target entity information to be in the state of the RL agent, and cannot be applied to tasks where the target entity is unknown. MINERVA [5] uses a policy gradient method to explore paths during training and test. Our proposed model further exploits state transition information by integrating the MCTS algorithm. Empirically, our proposed algorithm outperforms both DeepPath and MINERVA in the KBC benchmarks.[11]

## 6 Conclusion and Discussion

We developed an RL-agent (M-Walk) that learns to walk over a graph towards a desired target node for given input query and source nodes. Specifically, we proposed a novel neural architecture that encodes the state into a vector representation, and maps it to Q-values and a policy. To learn from sparse rewards, we propose a new reinforcement learning algorithm, which alternates between an MCTS trajectory-generation step and a policy-improvement step, to iteratively refine the policy. At test time, the learned networks are combined with MCTS to search for the target node. Experimental results on several benchmarks demonstrate that our method learns better policies than other baseline methods, including RL-based and traditional methods on KBC tasks. Furthermore, we also performed extensive experimental analysis to understand M-Walk. We found that our method is more accurate when the ground truth is in the candidate set. We also found that the out-of-candidate-set error is the main type of error made by M-Walk. Therefore, in future work, we intend to improve this method by reducing such out-of-candidate-set errors.

## Acknowledgments

We thank Ricky Loynd, Adith Swaminathan, and anonymous reviewers for their valuable feedback.

## Footnotes

*Yelong Shen, Jianshu Chen, and Po-Sen Huang contributed equally to the paper. The work was done when Yelong Shen and Jianshu Chen were with Microsoft Research. ⋆Po-Sen Huang is now at DeepMind (Email: posenhuang@google.com).

[2]Whenever the agent takes an action, by selecting an edge connected to a next node, the identity of the next node (which the environment will transition to) is already known. Details can be found in Section 2.

[3]The code of this paper is available at: https://github.com/yelongshen/GraphWalk

[4]An alternative choice is applying softmax to the Q-function to get the policy, which is known as softmax selection [27]. We found in our experiments that these two designs do not differ much in performance.

[5]There could be alternative ways to compute the score, such as $\text{Score}(n) = \max_{s_T \to n} Q_\theta(s_T, \text{STOP})$. However, we found in our (unreported) experiments that they do not make much difference.

[6]For simplicity, we use the same notation $q_t$ to denote its vector representation.

[7]We use these metrics in order to be consistent with [38, 5]. We also report the HITS and MRR scores for NELL995 in Table 9 of the supplementary material.

[8]ConvE: https://github.com/TimDettmers/ConvE, Neural-LP: https://github.com/fanyangxyz/Neural-LP/, DeepPath: https://github.com/xwhan/DeepPath, MINERVA: https://github.com/shehzaadzd/MINERVA/

[9]The ground truth is always in the candidate list in ConvE, as it examines all the nodes.

[10]More examples can be found in Appendix C.2 of the supplementary material.

[11]A preliminary version of M-Walk with limited experiments was reported in the workshop paper [22].

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
