[Supplementary Material]

# A Derivation of the recursion for $q_t$

Recalling the definition $q_t \triangleq s_{t-1} \cup \{a_{t-1}, n_t\}$ and using the recursion (1), we have

$$q_{t+1} \overset{(a)}{=} s_t \cup \{a_t, n_{t+1}\}$$

$$\overset{(b)}{=} s_{t-1} \cup \{a_{t-1}, n_t, \mathcal{E}_{n_t}, \mathcal{N}_{n_t}\} \cup \{a_t, n_{t+1}\}$$

$$\overset{(c)}{=} q_t \cup \{\mathcal{E}_{n_t}, \mathcal{N}_{n_t}, a_t, n_{t+1}\}$$

where step (a) uses the definition of $q_{t+1}$, step (b) substitutes the recursion (1), and step (c) uses the definition of $q_t$.

# B Algorithm Implementation Details

The detailed algorithm of M-Walk is described in Algorithm 1.

---
**Algorithm 1** M-Walk Training Algorithm

---
1: **Input:** Graph $\mathcal{G}$; Initial node $n_S$; Query $q$; Target node $n_T$; Maximum Path Length $T_{\max}$; MCTS Search Number $E$;
2: **for** episode $e$ in $[1..E]$ **do**
3:     Set current node $n_0 = n_S$; $q_0 = f_{\theta_q}(q, 0, 0, n_0)$
4:     **for** $t = 0 \ldots T_{\max}$ **do**
5:         Lookup from dictionary to obtain $W(s_t, a)$ and $N(s_t, a)$
6:         Select the action $a_t$ with the maximum PUCT value:

$$a_t = \mathrm{argmax}_a \left\{ c \cdot \pi_\theta(a|s_t)^\beta \frac{\sqrt{\sum_{a'} N(s_t, a')}}{1 + N(s_t, a)} + \frac{W(s_t, a)}{N(s_t, a)} \right\}$$

7:         Update $q_{t+1} = f_{\theta_q}(q_t, h_{A,t}, h_{a_t,t}, n_{t+1})$
8:         **if** $a_t$ is STOP **then**
9:             Compute estimated reward value $V_\theta(s_t) = Q(s_t, a_t = \texttt{STOP})$
10:             Add generated path $p$ into a path list
11:             Backup along the path $p$ to update the visit count $N(s_t, a)$ using (5) and the total action reward $W(s_t, a)$ using (6) on the $(s_t, a)$-th edge on the MCTS tree
12:             **Break**
13:         **end if**
14:     **end for**
15: **end for**
16: **for** each path $p$ in the path list **do**
17:     Set reward $r = 1$ if the end of the path $n_t = n_T$ otherwise $r = 0$
18:     Repeatedly update the model parameters with Q-learning:

$$\theta \leftarrow \theta + \alpha \cdot \nabla_\theta Q_\theta(s_t, a_t) \times \left( r(s_t, a_t) + \gamma \max_{a'} Q_\theta(s_{t+1}, a') - Q_\theta(s_t, a_t) \right)$$

19: **end for**

---

## B.1 MCTS implementation

In the MCTS implementation, we maintain a lookup table to record values $W(s_t, a)$ and $N(s_t, a)$ for each visited state-action pair. The state $s_t$ in the graph walk problem contains all the information along the traversal path, and $n_t$ is the node at the current step $t$. We assign an index $i_a$ to each candidate action $a$ from $n_t$, indicating that $a$ is the $i_a$-th action of the node $n_t$. Thus, the state $s_t$ can be encoded as a path string $P_{s_t} = (q, n_0, i_{a_0}, n_1, i_{a_1}, \ldots, n_t)$. We build a dictionary $\mathcal{D}$ using the path string as a key, and we record $W(s_t, a)$ and $N(s_t, a)$ as values in $\mathcal{D}$. In the backup stage, the $W(s_t, a)$ and $N(s_t, a)$ values are updated for each state-action pair along with the traversal path in

MCTS:

$$N(s_t, a) = N(s_t, a) + \gamma^{T-t} \tag{5}$$

$$W(s_t, a) = W(s_t, a) + \gamma^{T-t} V_\theta(s_T), \tag{6}$$

where $T$ is the length of the traversal path, $\gamma$ is the discount factor of the MDP, and $V_\theta(s_T)$ is the terminal state-value function modeled by $V_\theta(s_T) \triangleq Q(s_T, a = \mathtt{STOP})$.

In our experiments, the softmax temperature parameter $\tau$ in the policy network $\pi_\theta$ (see (3)) is set to be a constant. An alternative choice is to anneal it during training (e.g., $\tau = 1 \to 0$). However, we did not observe this to produce any significant difference in performance in our experiments. We believe the main reason is that $\pi_\theta$ is only used as a prior to bias the MCTS search, while the exploration of MCTS is controlled by the parameters $c$ and $\beta$ of (4).

## B.2 Experiment details

### B.2.1 Three Glass Puzzle

Figure 6: Graph traversal in the Three Glass Puzzle problem.

**An example**  Figure 6 illustrates one step in solving a Three Glass Puzzle. The following action sequences provide one solution to achieve the target $q = 4$, given initially empty containers with capacities $(A = 8, B = 5, C = 3)$, where $a, b, c$ denote the current contents of the containers:

- Initial state $\to (a = 0, b = 0, c = 0)$
- Fill $\mathcal{B} \to (a = 0, b = 5, c = 0)$
- Pour from $\mathcal{B}$ to $\mathcal{C} \to (a = 0, b = 2, c = 3)$
- Empty $\mathcal{C} \to (a = 0, b = 2, c = 0)$
- Pour from $\mathcal{B}$ to $\mathcal{C} \to (a = 0, b = 0, c = 2)$
- Fill $\mathcal{B} \to (a = 0, b = 5, c = 2)$
- Pour from $\mathcal{B}$ to $\mathcal{C} \to (a = 0, b = 4, c = 3)$

**Data generation**  In the Three Glass Puzzle experiments, we randomly draw four integers from $[1, 50)$ to represent the capacities $A, B, C$, and the desired volume $q$. We further restrict the values so that $A \geq B \geq C$ and $q < A$, to avoid data duplication. We discard puzzles for which there is no solution. Finally, we keep 600 unique puzzles as the experimental dataset, where 500 puzzles are used for training and the other 100 are used to test a model's generalization capability on the unseen test set.

**Experiment settings and hyperparameters**  Let $a, b, c$ be the current status of each container, and define the puzzle status at step $t$ as $n_t = [I_A^T, I_B^T, I_C^T, I_a^T, I_b^T, I_c^T]^T$, where $I_x$ is the one-hot representation to encode the value of $x$. Given that $A, B, C, a, b$ and $c$ are all smaller than 50 in the experiment, the dimension of $n_t$ is 300. The initial query $q$ is obtained by $q = E_{mb}[q]$, where $E_{mb}$ is a query embedding lookup table and $E_{mb}[x]$ indicates the $x$-th column. The query embedding dimension is set to 64. In the Three Glass Puzzle, there are 13 actions in total: fill one container to its capacity, empty one container, pour one container into another container, and a STOP action to terminate the game. We set the maximum length of an action sequence (i.e., the search horizon) to be 12, where only the STOP action can be taken on the final step. After the STOP action has been taken, the system evaluates the action sequence and assigns a reward $r = 1$ if the final status is a success, otherwise $r = 0$. The $f_{\theta_S}$ and $f_{\theta_A}$ functions are modeled by two different DNNs with the same architecture: two fully-connected layers with 32 hidden dimensions and ReLU activation function.

Table 5: A List of actions for each container in the Three Glass Puzzle. The agent can also determine to take the STOP action to terminate the game.

| Empty $\mathcal{A}$ | Fill $\mathcal{A}$ | Pour $\mathcal{A}$ to $\mathcal{B}$ | Pour $\mathcal{A}$ to $\mathcal{C}$ |
|---|---|---|---|
| Empty $\mathcal{B}$ | Fill $\mathcal{B}$ | Pour $\mathcal{B}$ to $\mathcal{A}$ | Pour $\mathcal{B}$ to $\mathcal{C}$ |
| Empty $\mathcal{C}$ | Fill $\mathcal{C}$ | Pour $\mathcal{C}$ to $\mathcal{A}$ | Pour $\mathcal{C}$ to $\mathcal{B}$ |

Table 6: Knowledge base completion datasets statistics.

| Dataset | # Train | # Test | # Relation | # Entity | avg. degree | median degree |
|---|---|---|---|---|---|---|
| WN18RR | 86,835 | 3,134 | 11 | 40,943 | 2.19 | 2 |
| NELL-995 | 154,213 | 3,992 | 200 | 75,492 | 4.07 | 1 |
| FB15K-237 | 272,115 | 20,466 | 237 | 14,541 | 19.74 | 14 |

(a) Test Beam / Rollout = 128    (b) Test Beam / Rollout = 300

Figure 7: Three Glass Puzzle test accuracy, where "PG" stands for policy gradient.

$f_{\theta_v}$ is two fully-connected layers with 16 hidden dimensions, where the first hidden layer uses a ReLU activation function and the output layer uses a linear activation function. $f_{\theta_q}$ is modeled by a GRU with hidden size 64. The hyperparameters in PUCT are set to $c = 0.5$ and $\beta = 0.2$. We use the ADAM optimization algorithm with learning rate 0.0005 during training, and we set the mini-batch size to 8.

### B.2.2 Knowledge Base Completion

**Statistics of the three datasets** The NELL-995 knowledge dataset contains $75,492$ unique entities and 200 relations. WN18RR contains $93,003$ triples with $40,943$ entities and 11 relations. And FB15k-237, a subset of FB15k where inverse relations are removed, contains $14,541$ entities and 237 relations. The detailed statstics are shown in Table 6.

**Experiment settings and hyperparameters** For the proposed M-Walk, we set the entity embedding dimension to 4 and relation embedding dimension to 64. The maximum length of the graph walking path (i.e., the search horizon) is 8 in the NELL-995 dataset and 5 in the WN18RR dataset. After the STOP action has been taken, the system evaluates the action sequence and assigns a reward $r = 1$ if the agent reaches the target node, otherwise $r = 0$. The initial query $q$ is the concatenation of the entity embedding vector and the relation embedding vector. The $f_{\theta_S}$ and $f_{\theta_A}$ functions are modeled by two different DNNs with the same architecture: two fully-connected layers with 64 hidden dimensions and the ReLU activation function. $f_{\theta_v}$ is two fully-connected layers with 16 hidden dimensions, where the first hidden layer uses a Tanh activation function and the output layer uses a linear activation function. $f_{\theta_q}$ is modeled by a GRU with hidden size 64. The hyperparameters in PUCT are set to $c = 2$ and $\beta = 0.5$. We roll out 32 MCTS paths in both training and testing in the NELL-995 dataset and 128 MCTS paths in the WN18RR dataset. We use the ADAM optimization algorithm for model training with learning rate 0.0001, and we set the mini-batch size to 8.

Table 7: Three Glass Puzzle test accuracy (%), where "Beam" denotes beam search.

| Size | 1 | 10 | 50 | 100 | 200 | 300 | 400 |
|---|---|---|---|---|---|---|---|
| PG (Beam) | 9.3 (2.1) | 30.7 (4.5) | 39.3 (3.2) | 45.3 (4.5) | 47.7 (3.2) | 48.7 (3.2) | 49.0 (2.6) |
| M-Walk (Beam) | 18.0 (1.7) | 46.0 (7.0) | 60.3 (7.8) | 67.0 (7.0) | 69.0 (6.2) | 69.3 (6.4) | 71.7 (4.5) |
| M-Walk (MCTS) | **18.0** (1.7) | **63.3** (5.0) | **84.3** (3.1) | **90.7** (2.5) | **95.0** (2.6) | **96.3** (1.5) | **99.0** (1.0) |

Table 8: BFS, DFS and M-Walk on Three Glass Puzzle.

| Method | Average # Steps | Max # Steps |
|---|---|---|
| BFS | 264.7 | 1030 |
| DFS | 192.2 | 1453 |
| M-Walk | 94.9 | 897 |

## C  Additional Experiments

### C.1  The Three Glass Puzzle task in different settings

We now present more experiments on the Three Glass Puzzle task under different settings. First, to see how fast M-Walk converges, we show in Figure 7 the learning curves of M-Walk and PG. It shows that M-Walk converges much faster than PG and achieves better results on this task. In Table 7, we report the test accuracy of M-Walk and vanilla policy gradient (REINFORCE/PG) with different beam search sizes and different MCTS rollouts during testing. The number of MCTS simulations for training M-Walk is fixed to be 32. We observe that M-Walk with MCTS achieves the best test accuracy overall. In addition, with larger beam search sizes and MCTS rollouts, the test accuracy improves substantially. Furthermore, replacing the MCTS in M-Walk by beam search at test time degrades the performance greatly, which shows that MCTS is also very important for M-Walk at test time.

As mentioned earlier, conventional graph traversal algorithms such as Breadth-First Search (BFS) and Depth-First Search (DFS) cannot be applied to the graph walking problem, because the ground truth target node is not known at test time. However, to understand how quickly M-Walk with MCTS can find the correct target node, we compare it with BFS and DFS in the following cheating setup. Specifically, we apply BFS and DFS to the test set of the Three Glass Puzzle task by disclosing the target node to them. In Table 8, we report the average traversal steps and maximum steps to reach the target node. The M-Walk with MCTS algorithm is able to find the target node more efficiently than BFS or DFS.

### C.1.1  Knowledge Graph Link Prediction

In this section, we first provide additional experimental results for the NELL995 and WN18RR tasks to support our analysis. In Figure 8, we show the positive reward rate during training on the NELL995 task. And in Figure 9, we provide more hyperparameter analysis (search horizon and MCTS simulation number) and training-time analysis. Furthermore, in Table 9, we show the HITS@K and MRR results on NELL995.

In addition, we conduct further experiments on the FB15k-237 dataset [29], which is a subset of FB15k [2] with inverse relations being removed. We use the same data split and preprocessing protocol as in [6] for FB15k-237. The results are reported in Table 10. We observe that M-Walk outperforms the other RL-based method (MINERVA). However, it is still worse than the embedding-based methods. In future work, we intend to combine the strength of embedding-based methods and our method to further improve the performance of M-Walk.

### C.2  The Reasoning (Traversal) Paths

In Table 11, we show the reasoning paths of M-Walk on the NELL995 dataset. Each reasoning path is generated by following the edges on the MCTS tree with the highest visiting count $N(s, a)$.

(a) TeamPlaySports     (b) AthletePlaysInLeague     (c) AthletePlaysSport

(d) OrganizationHeadquarterediInCity     (e) PersonBornInLocation     (f) AthleteHomeStadium

Figure 8: The positive reward rate during training (i.e., percentage of trajectories with positive reward during training) on the NELL-995 task.

(a) HITS@3     (b) HITS@10     (c) Training time

Figure 9: M-Walk hyperparameter and error analysis on WN18RR.

Table 9: The HITS@K and MRR results on the NELL995 dataset.

| Metric (%) | M-Walk | PG-Walk | Q-Walk | MINERVA | ComplEx | ConvE | DistMult |
|---|---|---|---|---|---|---|---|
| HITS@1 | **68.4** | 66.7 | 66.8 | 66.3 | 61.2 | 67.2 | 61.0 |
| HITS@3 | **81.0** | 77.5 | 77.3 | 77.3 | 76.1 | 80.8 | 73.3 |
| MRR | **75.4** | 74.8 | 74.5 | 72.5 | 69.4 | 74.7 | 68.0 |

Table 10: The results on the FB15k-237 dataset, in the form of "mean (standard deviation)".

| Metric (%) | M-Walk | PG-Walk | Q-Walk | MINERVA | ComplEx | ConvE | DistMult | NeuralLP |
|---|---|---|---|---|---|---|---|---|
| HITS@1 | 16.5(0.3) | 14.8(0.2) | 15.5(0.2) | 14.1(0.2) | 20.8(0.2) | 23.3(0.4) | 20.6(0.4) | 18.2(0.6) |
| HITS@3 | 24.3(0.2) | 23.3(0.3) | 23.8(0.4) | 23.2(0.4) | 32.6(0.5) | 33.8(0.3) | 31.8(0.2) | 27.2(0.3) |
| MRR | 23.2(0.2) | 21.3(0.1) | 21.8(0.2) | 20.5(0.3) | 29.6(0.2) | 30.8(0.2) | 29.0(0.2) | 24.9(0.2) |

Table 11: Examples of paths found by M-Walk on the NELL-995 dataset.

---

(i) **WorksFor:**

journalist jerome holtzman $\xrightarrow{\text{WorksFor}}$ ?

journalist jerome holtzman $\xrightarrow{\text{JournalistWritesForPublication}}$ website chicago tribune, (True)

politician mufi hannemann $\xrightarrow{\text{WorksFor}}$ ?

politician mufi hannemann $\xrightarrow{\text{PersonHasResidenceInGeopoliticalLocation}}$ city honolulu, (True)

ceo kumar birla $\xrightarrow{\text{WorksFor}}$ ?

ceo kumar birla $\xrightarrow{\text{PersonLeadsOrganization}}$ company hindalco, (True)

professor chad deaton $\xrightarrow{\text{WorksFor}}$ ?

professor chad deaton $\xrightarrow{\text{PersonLeadsOrganization}}$ BiotechCompany baker hughes, (False)

---

(ii) **TeamPlaySport:**

SportsTeam arizona dismond backs $\xrightarrow{\text{TeamPlaySport}}$ ?

SportsTeam arizona dismond backs $\xrightarrow{\text{TeamHomeStadium}}$ StadiumOrEventVenue chase field $\xrightarrow{\text{SportUsesStadium}^{-1}}$ sport baseball, (True)

SportsTeam l_a kings $\xrightarrow{\text{TeamPlaySport}}$ ?

SportsTeam l_a kings $\xrightarrow{\text{TeamPlaysAgainstTeam}^{-1}}$ SportsTeam red wings $\xrightarrow{\text{TeamWonTrophy}}$ AwardTrophyTournament stanley cup $\xrightarrow{\text{ChampionshipGameOfTheNationalSport}}$ sport hockey, (True)

SportsTeam cleveland browns $\xrightarrow{\text{TeamPlaySport}}$ ?

SportsTeam cleveland browns $\xrightarrow{\text{TeamPlaysAgainsTteam}^{-1}}$ SportsTeam yankees $\xrightarrow{\text{TeamHomeStadium}}$ AwardTrophyTournament yankee stadium $\xrightarrow{\text{SportUsesStadium}^{-1}}$ sport baseball, (False)

---