[Reviews · NeurIPS 2018]

Reviewer 1



This work tackles the problem of knowledge base completion (KBC) by walking through a knowledge graph. The graph walk is treated as an MDP with a known state transition model. MCTS is used to plan a trajectory through the graph, and these trajectories are used to train a value function with Q-learning. The Q-function is used to form a policy prior for the MCTS. A KBC-specific RNN structure is also proposed. Results on a range of KBC benchmarks show small but consistent gains over several baselines, including on-policy RL approaches and non-MCTS ablations. Quality MCTS is a strong planning algorithm for delayed-reward problems and seems a good fit for KBC, so this is an at the least an interesting application paper. The authors also extend previous work on combining MCTS with learned value functions and policy priors by leveraging the MCTS trajectories to learn the optimal value function of the KBC graph-traversal policy with Q-learning. This seems like a reasonable approach to improve the sample efficiency of learning the prior, compared to just updating the policy prior to match the MCTS visitation frequencies at the root node. The experiments seem to present competitive baselines and show some promise on a number of tasks. However, I would appreciate some more discussion or clarity of the exact roles of pi, Q, and V. Firstly, the background section 2 does not make the import distinction between an arbitrary policy and the optimal policy (and their corresponding value functions). In particular, note that Q-learning does not estimate Q^pi but Q* for the MDP. In M-Walk, Q is trained off-policy, and so learns about the optimal state-action value function Q* for graph-traversal. However, V is trained on-policy (wrt the MCTS tree policy, not pi_theta), but only on terminal MCTS states (not terminal MDP states, as I understand it). Could the authors elaborate on this choice (as opposed to for example using V = max_a (Q) )? Meanwhile, pi is simply a soft transformation of Q into a probability distribution (the authors note that simply using a softmax works similarly to their final implementation). Traditionally, Q-learning with Boltzmann exploration as the behaviour policy would anneal its entropy (ie temperature) to converge to a good solution. I am concerned that a fixed temperature would restrict the ability of the algorithm to develop strong confidence in its prior policy, even if it were justified by the data. In this particular setting with strongly bounded rewards and sufficient MCTS samples to correct for even an uncertain prior, it seems to work fine, but the exact mechanism by which the Q-learning transfers to the MCTS tree policy deserves some further discussion. Clarity The paper is quite clearly written on the whole. Some of the details of the architecture and intermediate state representation are a little hard to follow, and some design decisions regarding the neural architecture (s3.1) are a bit opaque. It should also be emphasised that the policy pi_theta is neither used to traverse the MDP (except indirectly as a prior) nor is its value learned by any function, which is a departure from the expectations set by section 2. Originality The combination of MCTS with learned policy and value priors has been explored in a number of domains, but this application to KBC appears novel and includes a number of algorithmic and architectural innovations. Significance This work addresses a significant and well-studied problem, as well as a significant class of solutions (MCTS with learned value/policy priors) with methods that may transfer to other problem settings. The experimental results are promising. However, more care with the formulation and explanation of the approach wrt the exact roles of the learned value functions and consequent policy prior would be very useful for solidifying the theoretical foundations of the approach.

Reviewer 2



In this paper, authors develop a Reinforcement Learning agent - M-Walk - that, given a query, learns to walk over a graph towards a desired target (answer) node. The proposed architecture maps the state into a vector representation, and maps it into a Q-network (for the policy) and a value network. The proposed RL algorithm alternates between MCTS trajectory generation steps, and policy improvement steps, for iteratively refining the policy. Question - other than WN18RR, FB15k-237 is also a great link prediction benchmark. Why didn't you consider it in your experiments? Is it because the number of relation types in FB15k-237 is much larger compared to WN18RR, causing a significant increase in the size of the action space? It would be great to have a brief discussion on this aspect in the paper, rather than omitting it completely.

Reviewer 3



Summary of the paper: - This paper proposes a walk-based method for knowledge graph completion (KGC). - For triples (s, q, o), an agent is trained so that given s and q, it traverses a path from the initial node s to the terminal node t in a knowledge graph. - This problem is formulated as that of Q-learning using RNN to model policy and Q functions, but the issue is the sparseness of rewards. - Monte Carlo tree search is used to overcome this issue. Pros: - The paper is clearly written. - The approach is natural and sound. - Very good performance compared with existing KGC methods. - Ablation study with PG-Walk and Q-Walk is nice. Cons: - The work is mostly empirical, without much theoretical justification (admittedly, as in many deep learning papers). - Not much novelty or surprise, given the success of AlphaGo. This is a type of paper in which a successful technique in other fields is applied to a different field. I could not see exactly what is the novelty/creativity in terms of how MCTS is applied for this specific problem of KGC. - No analysis on running time is provided. General Comments: This paper is well written (but I would like the content of Section 4.2 to be more organized --- Page 7 consists of a single paragraph now!). The novelty of the proposal is somewhat questionable, because reinforcement learning has already been introduced for KGC, and the application of MCTS naturally comes to mind. Although this work is mostly empirical and little theoretical contribution is provided, it still achieves the state-of-the-art performance, and that could be a merit for its publication. Questions: * Related work MCTS has been introduced in walk-based KGC by Sheng et al. in ICLR 2018 Workshop (ReinforceWalk: Learning to Walk in Graph with Monte Carlo Tree Search), and this work must be cited. Please elaborate on how M-Walk differs from ReinforceWalk as much as you can tell from the extended abstract. * Experiments: - Dataset: Is there any reason the comparison is done with NELL995 and WN18RR? FB15k is also the standard dataset. - Baselines Why do the set of compared methods differ between datasets? In particular, we know that TransE and TransR are somewhat weaker compared with more recent bilinear models, such as ComplEx, ANALOGY, and ConvE. - Evaluation metrics Also, why is the MAP score used on NELL995 whereas HITS@/MRR are used on WN18RR? I have nothing against MAP scores, but typically HITS@/MRR are reported in literature on KGC even on the NELL-995 dataset (e.g., the MINERVA paper); using different metrics makes it difficult to validate the experimental results. - Figure 6(a) Is there any explanation as to why accuracy drops as the number of roll outs are increased to 256? * Application to classification task The method is only concerned with prediction of knowledge triple in which only one entity is missing. Can the proposed method be applied to the triple classification task where the truth of a given triple must be predicted? This is more or less straightforward in score-function-based models (provided that a suitable threshold can be determined), but I cannot see how this can be addressed with walk-based methods in general. * Running time Training and inference both involves roll-out, which raises concern about the running time of the algorithms. How does M-walk fare against scoring-based methods (e.g., ComplEx) or MINERVA in terms of running time? Please provide the time required for training and inference. I am curious about inference time in particular.